# mTOR-dependent phosphorylation controls TFEB nuclear export

Gennaro Napolitano[1,2], Alessandra Esposito[1], Heejun Choi [3], Maria Matarese[1], Valerio Benedetti[1], Chiara Di Malta[1], Jlenia Monfregola[1], Diego Luis Medina [1], Jennifer Lippincott-Schwartz[3,4] & Andrea Ballabio[1,2,5]

During starvation the transcriptional activation of catabolic processes is induced by the nuclear translocation and consequent activation of transcription factor EB (TFEB), a master modulator of autophagy and lysosomal biogenesis. However, how TFEB is inactivated upon nutrient refeeding is currently unknown. Here we show that TFEB subcellular localization is dynamically controlled by its continuous shuttling between the cytosol and the nucleus, with the nuclear export representing a limiting step. TFEB nuclear export is mediated by CRM1 and is modulated by nutrient availability via mTOR-dependent hierarchical multisite phosphorylation of serines S142 and S138, which are localized in proximity of a nuclear export signal (NES). Our data on TFEB nucleo-cytoplasmic shuttling suggest an unpredicted role of mTOR in nuclear export.

[1] Telethon Institute of Genetics and Medicine (TIGEM), Via Campi Flegrei 34, 80078 Pozzuoli, Naples, Italy. [2] Medical Genetics Unit, Department of Medical and Translational Science, Federico II University, Via Pansini 5, 80131 Naples, Italy. [3] Howard Hughes Medical Institute, Janelia Research Campus, Ashburn, VA 20147, USA. [4] National Institute of Child Health and Development, National Institutes of Health, Bethesda, MD 20892, USA. [5] Department of Molecular and Human Genetics and Neurological Research Institute, Baylor College of Medicine, Houston, TX 77030, USA. Correspondence and requests for materials should be addressed to A.B. (email: ballabio@tigem.it)

Transcription factor EB (TFEB) is a member of the MiT-TFE helix–loop–helix leucine-zipper (bHLH-Zip) family of transcription factors and plays a pivotal role in organelle biogenesis and cell metabolism[1]. TFEB acts as a global controller of lysosomal biogenesis[2], autophagy[3], lysosomal exocytosis[4], lipid catabolism[5], energy metabolism[6], and also plays important roles in the modulation of the immune response[7]. As a master modulator of intracellular clearance pathways, TFEB has been used as a therapeutic tool in cellular and mouse models of diseases characterized by accumulation of toxic aggregates, including lysosomal storage disorders[4,8–10] and neurodegenerative diseases[11–16]. TFEB and other members of the MiT-TFE transcription factors are often deregulated in different types of cancer[1], suggesting that the pharmacological modulation of TFEB activity may represent a relevant therapeutic approach for a wide number of diseases.

The activity of TFEB is tightly linked to nutrient availability via protein phosphorylation. In the presence of nutrients TFEB is phosphorylated by mechanistic target of rapamycin (mTOR) on S142 and S211 serine residues, which play a crucial role in determining TFEB subcellular localization. When these serines are phosphorylated, TFEB is mainly cytosolic and inactive[17–19]. Recent studies showed that additional, mTOR-dependent (S122)[20] or -independent (S138 and S134)[21], phosphorylation sites play a role in the modulation of TFEB localization, indicating that other kinases may also regulate TFEB activity. Accordingly, ERK was also shown to contribute to S142 phosphorylation and modulation of TFEB subcellular localization[3]. Serine-to-alanine mutations of either S142, S138, or S211 induce constitutive nuclear localization and activity of TFEB[3,17–19,21]. Phosphorylation of S211 has been shown to serve as a recognition site for TFEB binding to 14–3–3 and cytosolic retention[17,18]; however, how S142 and S138 phosphorylation affects TFEB localization is currently unknown.

Upon starvation or lysosomal stress, inhibition of mTOR and concomitant activation of the phosphatase calcineurin by TRPML1-mediated lysosomal calcium release induces TFEB dephosphorylation. This results in a nuclear localization of TFEB[22]. However, the mechanism by which TFEB then redistributes from the nucleus to the cytosol upon nutrient refeeding is completely unknown.

Here we show that nutrients promote cytosolic relocalization of nuclear TFEB via CRM1-dependent nuclear export. We found that TFEB continuously shuttles between the cytosol and the nucleus and that the nutrient-dependent modulation of nuclear export rates plays a major role in controlling TFEB subcellular localization. Mechanistically, our data reveal the presence of a nuclear export signal (NES) localized in the N-terminal portion of TFEB, whose integrity is absolutely required for TFEB nuclear export. Remarkably, we found that nutrient- and mTOR-dependent hierarchical phosphorylation of S142 and S138, which are localized in proximity of the NES, is necessary to induce TFEB nuclear export and inactivation. Our data highlight a new mechanism controlling TFEB subcellular localization and activity via the modulation of nuclear export.

## Results

**TFEB continuously shuttles between the cytosol and the nucleus via CRM1-dependent nuclear export.** Starvation is known to induce calcineurin-mediated TFEB dephosphorylation and nuclear translocation[22], however how nutrient refeeding induces TFEB inactivation is currently unknown. Time-lapse imaging of cells stably expressing TFEB-GFP revealed that TFEB rapidly redistributes from the nucleus to the cytosol within 20 min upon nutrient replenishment (Fig. 1a), indicating that TFEB undergoes active nuclear export. Interestingly, a recent interactome analysis aimed at identifying binding partners of the major exportin CRM1 identified TFEB as a strong CRM1-interacting protein[23]. Consistently, we found that TFEB redistribution from the nucleus to the cytoplasm is mediated by active CRM1-mediated nuclear export, as it is impaired by the addition of leptomycin B, a known CRM1 inhibitor[24,25], in both HeLa (Fig. 1b, c) and HEK293T cells (Fig. 1d, e). Similar results were also obtained upon silencing of CRM1 (Supplementary Fig. 1A), which caused a severely impaired nuclear export of TFEB upon nutrient refeeding in HeLa cells (Fig. 1f, g). Similar results were also obtained in HEK293T and ARPE19 cells (Supplementary Fig. 1B, C)

Surprisingly, leptomycin B treatment in fed cells, in which TFEB has a predominant cytoplasmic localization[17–19], was sufficient to induce progressive accumulation of TFEB in the nuclear compartment (Fig. 2a), suggesting that even in fed cells TFEB continuously shuttles between the cytosol and the nucleus. These data were further corroborated by time-lapse imaging showing that TFEB rapidly accumulates in the nuclei of fed cells treated with leptomycin B, with the rates of nuclear accumulation comparable to those observed during starvation (Supplementary Fig. 1D).

Next, we performed fluorescence recovery after photobleaching (FRAP) experiments under different conditions to monitor changes in nuclear and cytosolic TFEB-GFP signal over time after photobleaching cytosolic TFEB-GFP (Fig. 2b–d and Supplementary Fig. 2). Significant redistribution of TFEB from nucleus to cytoplasm was seen under refeeding conditions (Fig. 2b–d), supporting the notion that nuclear export is an important modulator of TFEB subcellular localization. Notably, under starvation conditions, a fraction of nuclear TFEB still redistributed from nucleus to cytosol (Fig. 2b–d), further indicating that TFEB undergoes continuous nucleo-cytoplasmic shuttling.

Interestingly, TFEB nucleo-cytoplasmic shuttling during both starvation and refeeding was blocked by Torin treatment, with the rate of TFEB nuclear loss in the presence of Torin similar to that observed in the presence of leptomycin B (Fig. 2b–d). These results indicate that TFEB subcellular localization is much more dynamic than previously appreciated and that the net localization and optical detection of TFEB in either the nucleus or the cytoplasm is the result of the relative nuclear import and export rates, which are influenced by nutrient availability and mTOR activity.

**A nuclear export signal controls TFEB cytosolic redistribution.** The finding that TFEB undergoes CRM1-dependent nuclear export prompted us to search for the presence of a nuclear export signal (NES) in the TFEB protein. We found that TFEB contains a CRM1 consensus hydrophobic sequence located at the N-terminal portion of the protein that is highly evolutionary conserved (Fig. 3a) and that is also present in other MiT-TFE members, such as TFE3 and MITF (Fig. 3b). Strikingly, mutagenesis of three different hydrophobic residues within the putative NES, namely M144, L147, and I149, completely impaired cytosolic relocalization of TFEB upon refeeding, indicating that the integrity of the NES sequence is required for TFEB nuclear export (Fig. 3c, d). To further validate this point, we performed FRAP experiments to evaluate the nutrient-dependent cytosolic redistribution of NES-mutant TFEB-GFP. Remarkably, we found that the NES-mutants M144A, L147A, and I149A displayed severely impaired nuclear export kinetics (Fig. 3e, f and Supplementary Fig. 3). These data suggest that NES-mediated, CRM1-dependent nuclear export allows TFEB nuclear export in response to nutrient replenishment.

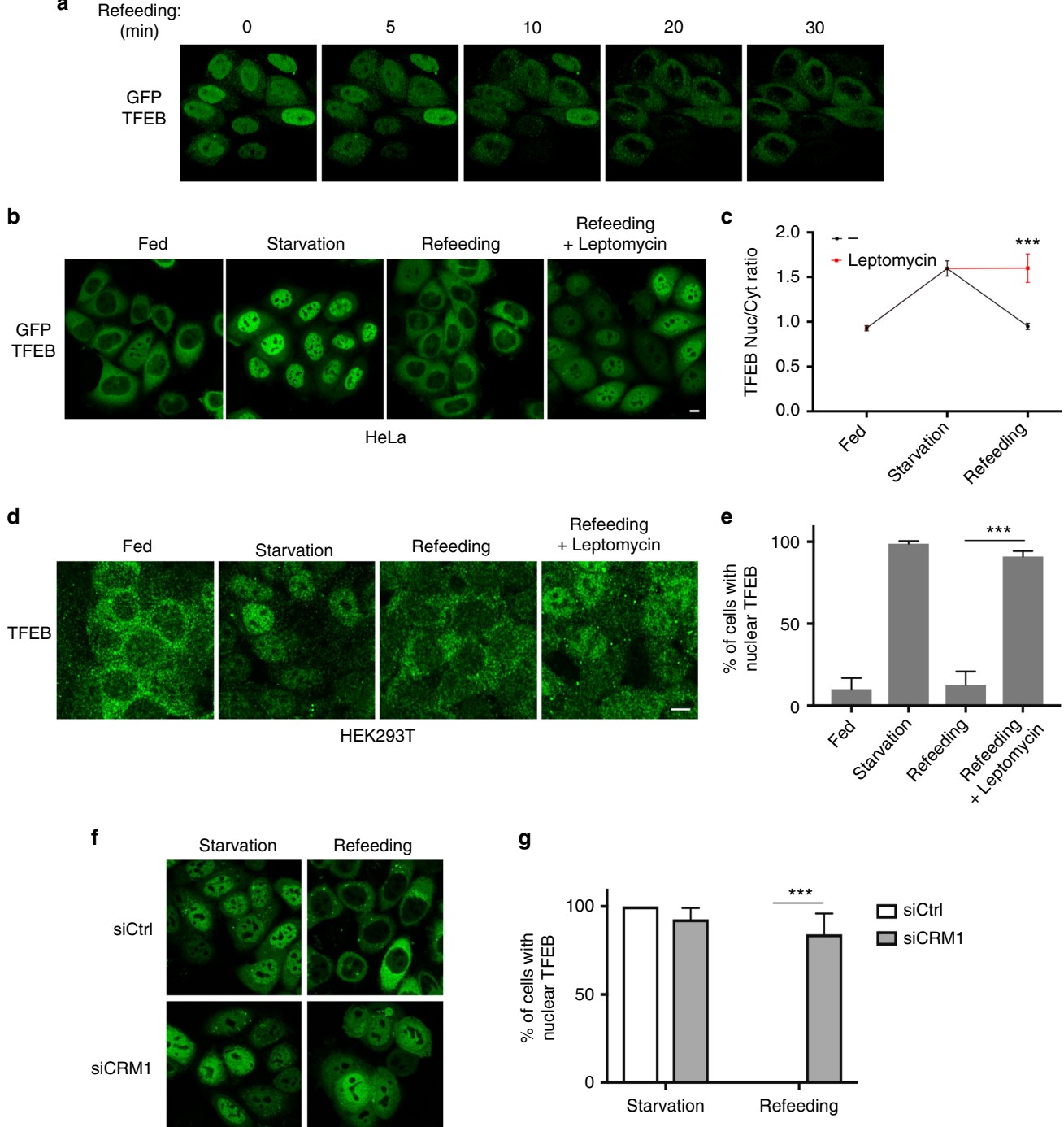

**Fig. 1** TFEB undergoes CRM1-dependent nuclear export. **a** Time-lapse analysis of HeLa cells expressing TFEB-GFP and restimulated with nutrients upon 60-min starvation. Each panel shows a 3D reconstruction of a selected frame at the indicated time points using the ImageJ software. **b** HeLa cells stably expressing TFEB-GFP were either left untreated (Fed), starved for amino acids for 60 min (starvation), or restimulated with amino acids for 30 min in the absence or in the presence of the CRM1 inhibitor leptomycin B (5 nM) and analyzed by confocal microscopy. **c** HeLa cells stably expressing TFEB-GFP were plated in 96-well plates and treated as in **b**, analyzed by automated high-content imaging and calculated for the ratio between nuclear and cytosolic TFEB fluorescence intensity as described in the "Methods" section. Each dot represents the average nucleo/cytoplasmic TFEB fluorescence intensity ratio analyzed in several hundred cells from three different wells. Results are mean ± SEM. $n > 1500$ cells per condition. ***$P < 0.001$, two-way ANOVA. **d** HEK293T cells were treated as in **b** and analyzed by confocal microscopy. **e** HEK293T shown in **d** were analyzed to calculate the percentage of cells showing nuclear TFEB localization. Results are mean ± SEM. $n > 120$ cells per condition. ***$P < 0.001$, unpaired $t$-test. **f** HeLa cells stably expressing TFEB-GFP and transfected with siRNA directed against CRM1 (siCRM1) or with control siRNA (siCtrl) were either starved for 60 min (starvation), or starved and restimulated with amino acids for 30 min and analyzed by confocal microscopy. **g** Cells shown in **f** were analyzed to calculate the percentage of cells showing nuclear TFEB localization. $n > 30$ cells per condition. Results are mean ± SEM. ***$P < 0.001$, two-way ANOVA. Scale bars: 10 μm

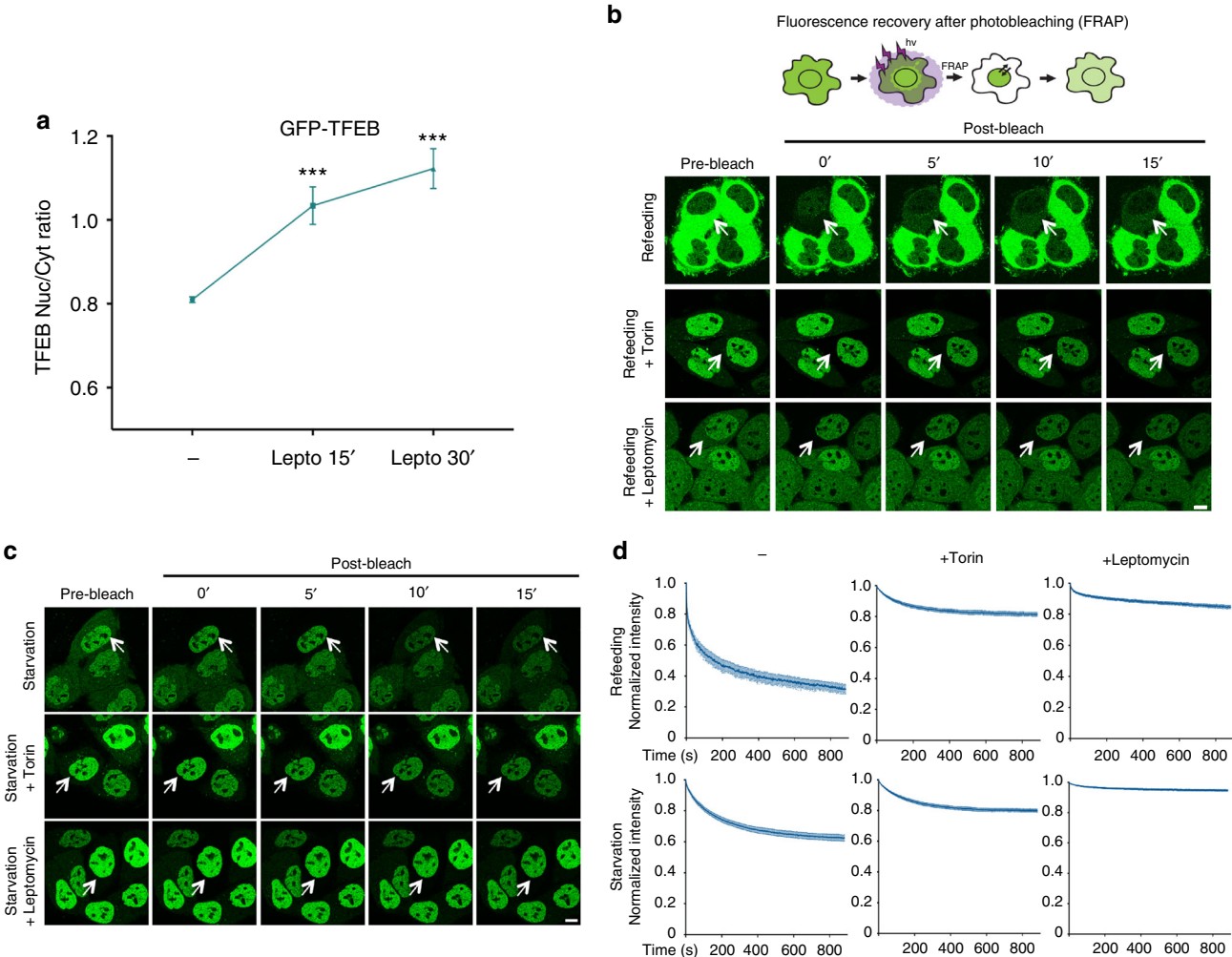

**Fig. 2** TFEB continuously shuttles between the cytosol and the nucleus. **a** HeLa cells stably expressing TFEB-GFP were treated with leptomycin B for the indicated time points and analyzed for TFEB subcellular localization by automated high-content imaging. Results are mean ± SEM. $n > 1200$ cells per condition. ***$P < 0.001$, one-way ANOVA. **b** Schematic experimental scheme and representative time-lapse images of TFEB-GFP-expressing HeLa cells treated as indicated and imaged for the indicated time points upon photobleaching of cytosolic TFEB. **c** TFEB-GFP-expressing HeLa cells were treated as indicated and imaged upon photobleaching of cytosolic TFEB as in **b**. **d** Cells described in **b** and **c** were analyzed and plotted for the decay of TFEB nuclear fluorescence using ImageJ software. $n > 6$ cells per condition. Results are mean ± SEM. Scale bars: 10 μm

**TFEB nuclear export kinetics are modulated by nutrient availability**. To directly assess the nuclear export kinetics of TFEB in response to nutrient availability, we performed fluorescence loss in photobleaching (FLIP) experiments by photobleaching cytosolic or nuclear TFEB (Fig. 3g, h). The loss of nuclear signal upon continuous photobleaching of cytosolic TFEB followed a bi-exponential fitting, suggesting the presence of two different nuclear pools of TFEB that are exported from the nucleus at different rates. The bi-exponential fitting revealed characteristic time constants of the two pools that were $\tau_{slow} = 216$ s and $\tau_{fast} = 70$ s. Interestingly, the time constants did not change during starvation and refeeding conditions, but the relative amounts of slow and fast populations did change. Under refeeding, 80% of nuclear TFEB comprised the fast population and 20% comprised the slow population. Under starvation, 35% of nuclear TFEB comprised the fast population and 65% the slow population (Fig. 3h). These results indicate that nutrients promote faster TFEB export kinetics and suggest that the two different TFEB pools being exported at different rates may correspond to the phosphorylated and dephosphorylated forms.

**TFEB nuclear export is controlled by mTOR-dependent hierarchical phosphorylation**. To test whether TFEB phosphorylation affects its nuclear export, we analyzed constitutively nuclear-mutated versions of TFEB in which specific serines, namely S142, S138, and S211, were mutated into alanines[17–19,21]. FRAP experiments in which we monitored nuclear export of TFEB after photobleaching of the cytoplasm revealed a dramatically impaired export of TFEB mutants S142A and S138A compared to wild-type TFEB (i.e., only 20 and 30% of the total redistributed to the cytosol 15 min after photobleaching for S142A and S138A, respectively—Fig. 4a, b). These results indicate that phosphorylation of TFEB at S142 and S138 sites is necessary for TFEB nuclear export. Interestingly, both S142 and S138 are located in close proximity of TFEB NES (Fig. 2a, b). By contrast, TFEB-S211A redistribution from nucleus to cytoplasm measured in the FRAP experiments was much faster than the S142A and S138A mutants and was only marginally impaired compared to wild-type TFEB (Fig. 4a, b).

Next, we used specific phosphoantibodies recognizing phospho-S142 and phospho-S138 TFEB residues, respectively. Analysis of TFEB phosphorylation revealed that these mutants

showed hierarchical phosphorylation patterns (Fig. 4c). In particular, TFEB S142A showed impaired phosphorylation of S142 and S138; TFEB S138A showed impaired phosphorylation of S138, but normal S142 phosphorylation; importantly, in TFEB S211A, which still retained the ability to be exported from the nucleus, S142 and S138 were normally phosphorylated (Fig. 4c).

These data suggest that phosphorylation of S142 acts as a priming site for subsequent S138 phosphorylation, and that phosphorylation on these residues is required for TFEB nuclear export.

In addition, our results indicate that nutrients may promote nuclear TFEB phosphorylation to allow its export. Accordingly, we found that TFEB was efficiently rephosphorylated on S142

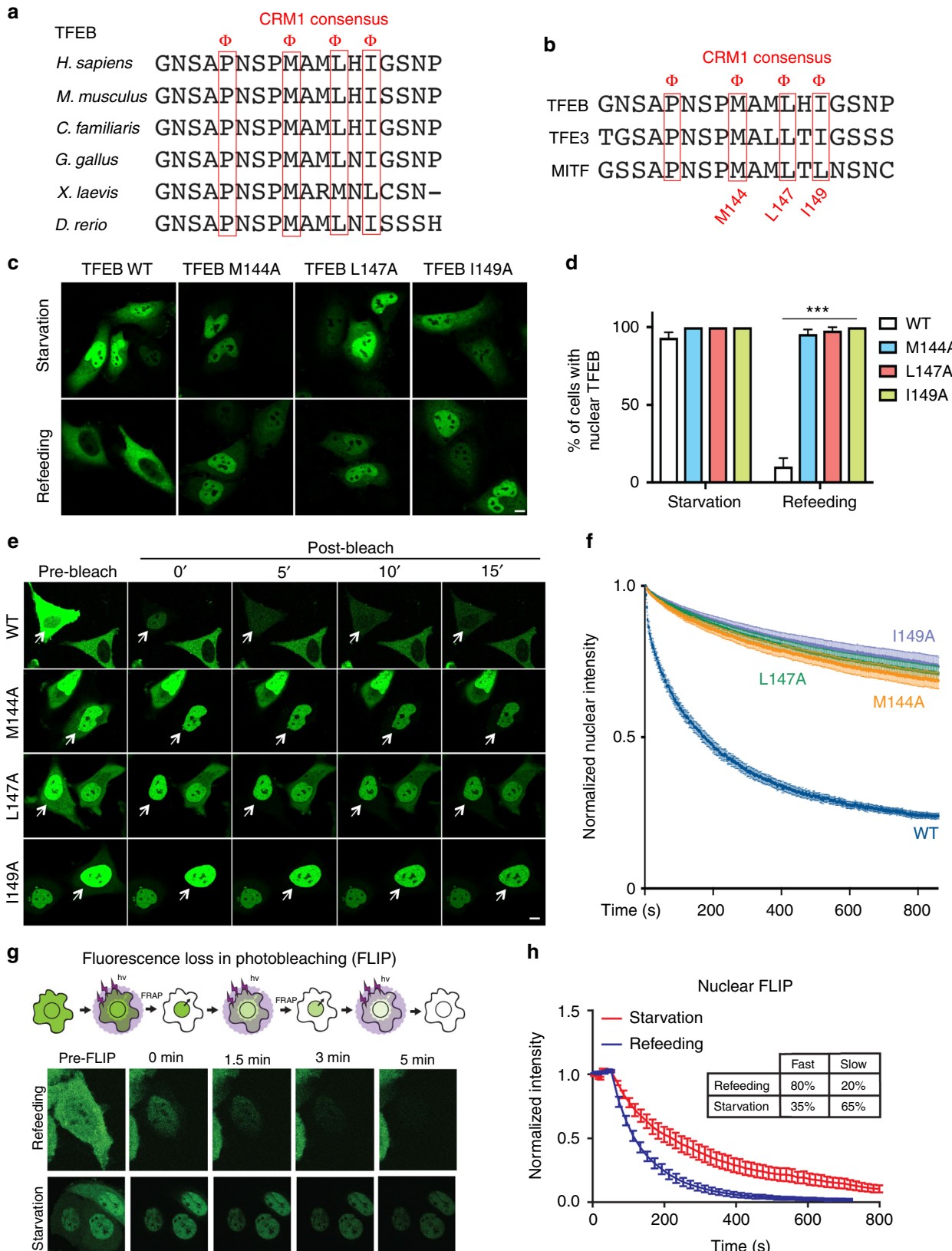

and S138 upon starvation/refeeding in the presence of leptomycin B (Fig. 5a), which caused an almost complete nuclear retention of TFEB (Fig. 5b). Importantly, TFEB phosphorylation was completely abolished by Torin treatment, indicating that mTOR activity is required for the phosphorylation of nuclear TFEB (Fig. 5a). Consistently, the NES-mutant M144A TFEB, which shows a nuclear localization in response to starvation/refeeding (Fig. 2), showed efficient nutrient-dependent phosphorylation of S142 and S138, further indicating that these residues are phosphorylated in the nuclear compartment (Fig. 5c). In addition, time course analysis of TFEB rephosphorylation upon refeeding showed that the phosphorylation of the M144A TFEB nuclear export mutant is highly efficient, and even enhanced, at early time points upon refeeding compared to WT TFEB (Supplementary Fig. 4A). Conversely, the phosphorylation of a mutated version of TFEB (ΔNLS-TFEB)[18], which lacks a nuclear localization signal and shows a constitutively cytosolic localization (Supplementary Fig. 4B), was markedly impaired compared to wild-type TFEB (Fig. 5d). Importantly, phosphorylation of both S142 and S138 was completely abolished by Torin treatment despite concomitant hyperactivation of ERK and GSK3 (Fig. 5e), which were shown to be able to phosphorylate TFEB on S142 and S138, respectively. In contrast, ERK and GSK3 inhibition had a limited effect on TFEB phosphorylation (Fig. 5e), indicating that mTOR has a predominant role in the modulation of TFEB.

Thus, our results suggest a differential role of specific serine residues in the control of TFEB nucleo-cytoplasmic shuttling and indicate that mTOR-dependent S142 and S138 nuclear phosphorylation is required for TFEB nuclear export.

## Discussion

Our data indicate that TFEB continuously shuttles between the cytosol and the nucleus, suggesting that TFEB activation is finely controlled by the relative net rates of nuclear import and export (Supplementary Fig. 5). Phosphorylation is known to influence the dynamics of nucleo-cytoplasmic shuttling[26]. In the case of TFEB, this mechanism ensures a timely and continuous tuning of TFEB activity based on its phosphorylation status. Our data based on leptomycin B treatment and CRM1 silencing indicate that nutrients regulate TFEB nuclear export in a CRM1-dependent manner. This is consistent with the results of a recent proteomic analysis aimed at the identification of all CRM1-binding proteins. Strikingly, TFEB ranked as the second highest-scoring protein, among more than a thousand CRM1-interacting proteins, for its ability to bind CRM1[23]. These data, together with the identification of a consensus hydrophobic NES sequence located at the N-terminal portion of TFEB, strongly suggest that TFEB is a bona fide cargo of CRM1-mediated nuclear export.

Our data show that nutrient availability regulates TFEB nucleo-cytoplasmic shuttling through hierarchical phosphorylation of specific serine residues. These serines have different and complementary roles in the regulation of TFEB subcellular localization. TFEB S142A and S138A mutants showed highly impaired

export kinetics, whereas the S211A mutant, which is efficiently phosphorylated on S142 and S138 (Fig. 4c), retained, by enlarge, its ability to be exported. Interestingly, the NES of TFEB encompasses the S142 phosphorylation site and is proximal to S138 (Fig. 2a, b). Phosphorylation is known to mediate the induction of CRM1-dependent nuclear export of a variety of proteins ranging from transcription factors to protein kinases[27–30]. Thus, it is possible that S142 and S138 phosphorylation is required for the recognition and binding of the TFEB NES by CRM1, which is crucial for efficient nuclear export, whereas phosphorylation of S211 may mediate cytosolic retention via 14–3–3 binding, as previously shown[17,18]. This mechanism ensures that only a fully phosphorylated TFEB is completely cytosolic and inactive, indicating that nutrient levels finely control TFEB subcellular localization via modulation of its shuttling kinetics (Supplementary Fig. 5).

Serine residue S142 has been shown to be a site for both ERK- and mTOR-mediated phosphorylation[3], whereas S138 has been proposed as a GSK3-phosphorylated site[21]. However, our data show that phosphorylation on both S142 and S138 entirely depends on mTOR activity (Fig. 5). In addition, Torin treatment induces complete TFEB dephosphorylation despite concomitant hyperactivation of both ERK and GSK3 (Fig. 5e), indicating that these kinases are not able to phosphorylate TFEB in the absence of mTOR activity. This suggests that mTOR-mediated phosphorylation is a predominant mechanism regulating TFEB subcellular localization.

Our data also suggest that mTOR may phosphorylate TFEB in the nuclear compartment. This hypothesis is supported by multiple pieces of evidence: first, TFEB is efficiently rephosphorylated upon refeeding in leptomycin-treated cells, in which TFEB shows a predominantly nuclear localization as a result of impaired nuclear export (Fig. 5a, b); second, the M144A TFEB mutant shows efficient rephosphorylation even at early time points upon refeeding (Fig. 5c and Supplementary Fig. 4A) despite its nuclear export being dramatically impaired (Fig. 3c–f); third, the constitutively nuclear S138A TFEB mutant shows efficient S142 phosphorylation (Fig. 4c); finally, deletion of the NLS results in impaired phosphorylation of a fully cytosolic TFEB (Fig. 5d and Supplementary Fig. 4B).

Multiple studies have shown that mTOR localizes and functions in the nuclear compartment[31–37]. Therefore, it is possible that a nuclear pool of mTOR is responsible for TFEB phosphorylation and induction of nuclear export. Interestingly, it has been proposed that NFAT is phosphorylated by mTOR in the nucleus and dephosphorylated in the cytoplasm through calcineurin[38]. This mechanism is strikingly similar to the one observed for TFEB. However, whether mTOR directly phosphorylates TFEB in the nucleus or if other kinases are involved in the modulation of TFEB nuclear export requires further investigation.

In summary, our data uncover a new mechanism by which nutrient availability controls TFEB localization and activity. This may unveil new strategies for pharmacological intervention in

**Fig. 3** The kinetics of TFEB nuclear export are modulated by nutrients via a nuclear export signal (NES). **a** Cross-species and **b** intra-family sequence alignment of a CMR1 consensus sequence located in the N-terminus of TFEB. Φ: hydrophobic residue. **c** HeLa cells were transfected with either WT or NES-mutant TFEB, subjected to starvation/refeeding and analyzed by confocal microscopy. **d** Cells described in **c** were analyzed to calculate the percentage of cells showing nuclear TFEB localization. $n > 20$ cells per condition. Results are mean ± SEM. ***$P < 0.001$, two-way ANOVA. **e** Representative time-lapse images of HeLa cells transfected with either wild-type TFEB-GFP or with TFEB mutants M144A, L147A, and I149A. Twenty-four hours after transfection, cells were imaged for the indicated time points upon photobleaching of cytosolic TFEB in FRAP experiments. **f** Cells described in **e** were analyzed and plotted for the decay in TFEB nuclear fluorescence. Results are mean ± SEM. $n > 12$ cells per condition. **g** Schematic experimental scheme and representative time-lapse images of HeLa cells stably expressing TFEB-GFP, treated as indicated and imaged for the indicated time points in FLIP experiments. **h** Cells described in **g** were analyzed and plotted for the decay in TFEB nuclear fluorescence. Results are mean ± SEM. Scale bars: 10 μm

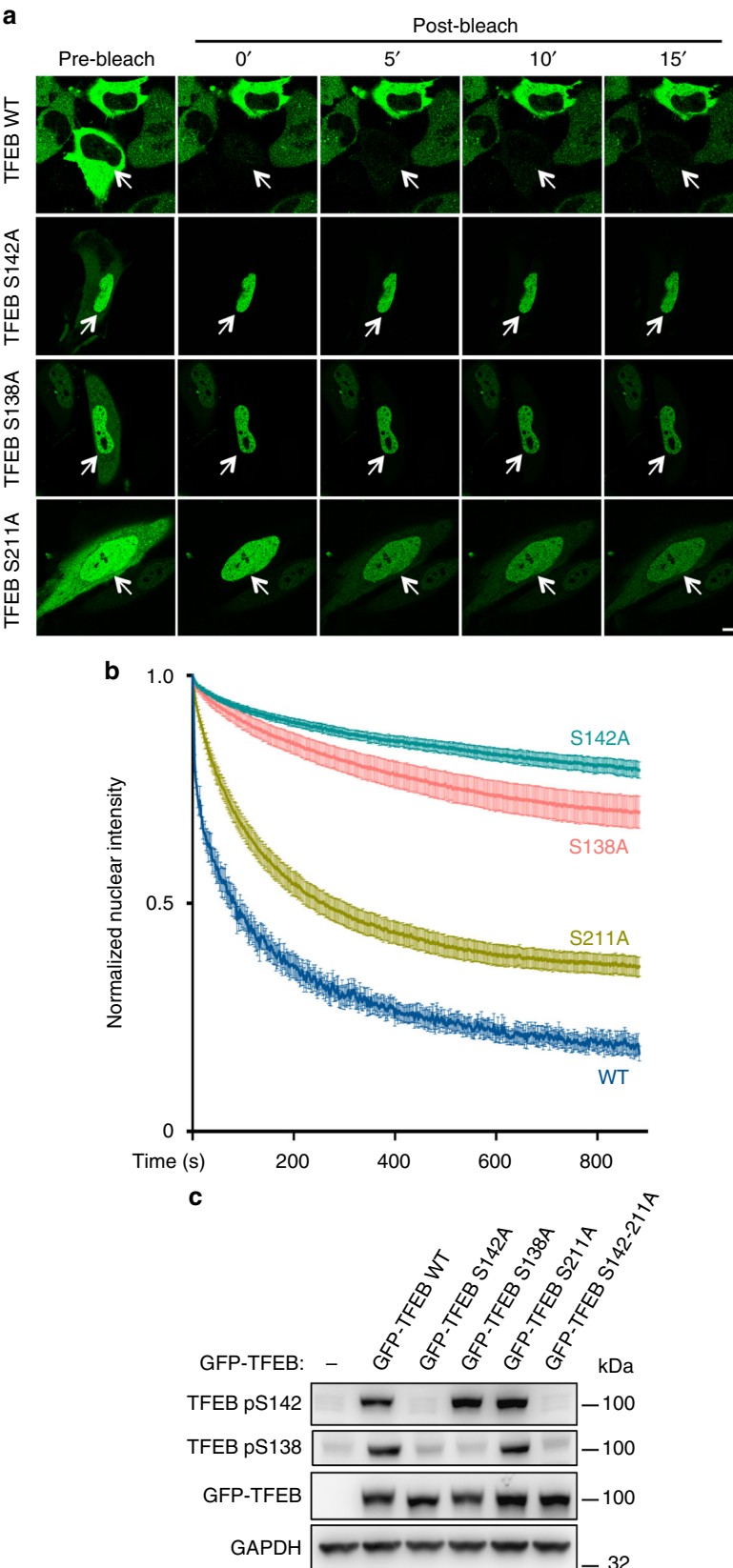

**Fig. 4** Hierarchical phosphorylation controls TFEB nuclear export. **a** Representative time-lapse images of HeLa cells transfected with either wild-type TFEB-GFP or with TFEB mutants S142A, S138A, and S211A. Twenty-four hours after transfection, cells were subjected to FRAP experiments. Scale bar: 10 μm. **b** Cells described in **a** were analyzed and plotted for the decay in TFEB nuclear fluorescence. Results are mean ± SEM. $n > 7$ cells per condition. **c** HeLa cells were transfected with either wild-type or serine-to-alanine mutant TFEB for 24 h and evaluated for TFEB phosphorylation by immunoblotting using the indicated antibodies

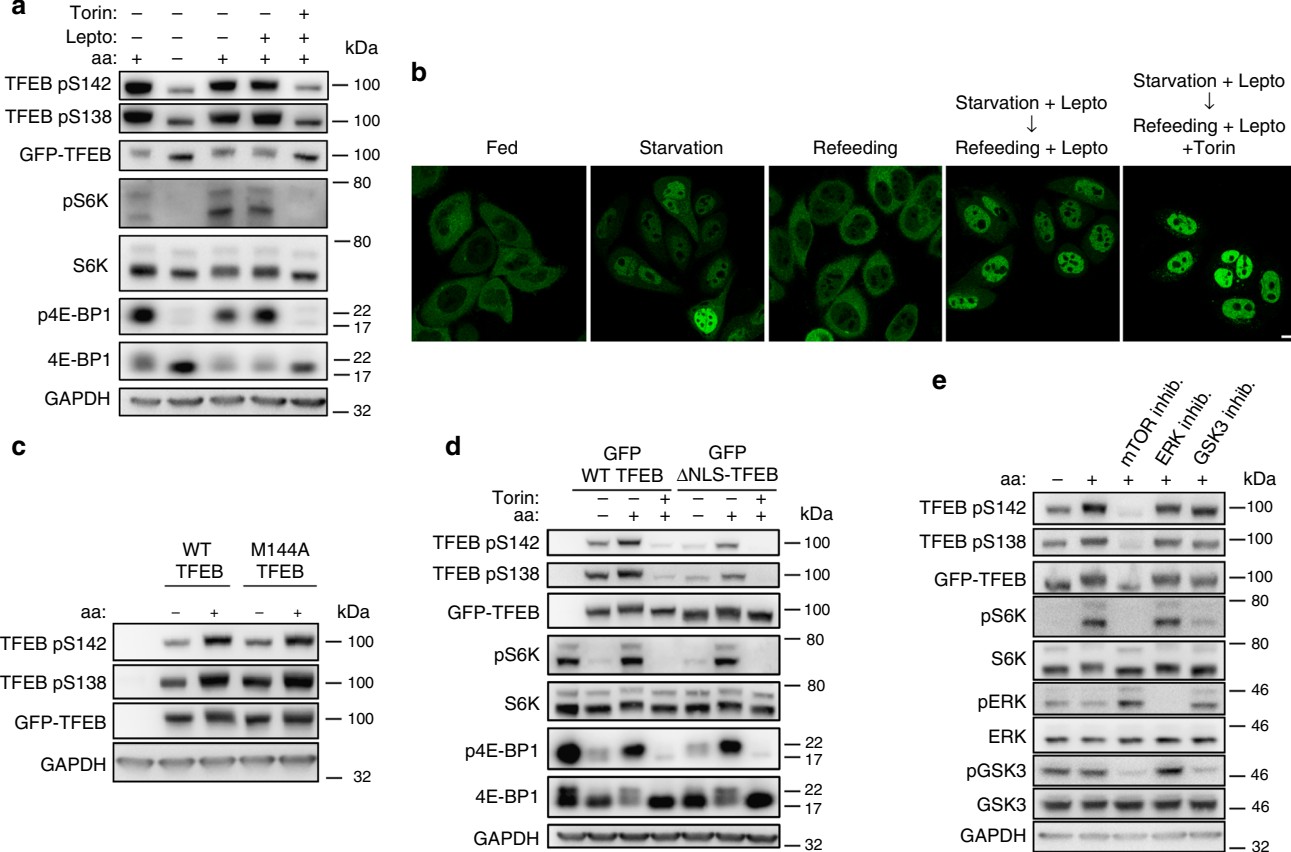

**Fig. 5** TFEB undergoes mTOR-dependent nuclear phosphorylation. **a** HeLa cells stably expressing TFEB-GFP were either starved of amino acids (aa) for 60 min, or starved and restimulated with amino acids for 30 min. Where indicated, cells were pretreated with 5 nM leptomycin B (Lepto) during starvation, prior to refeeding in the presence or absence of leptomycin B or Torin (250 nM). Upon treatment, cell extracts were analyzed by western blotting with the indicated antibodies. **b** Representative images of cells treated as in **a** and analyzed for TFEB localization by confocal microscopy. Scale bar: 10 µm. **c** HeLa cells were transfected with wild-type or M144A TFEB for 24 h. Cells were then either starved for 1 h or starved and restimulated with nutrients for 30′. Cell extracts were analyzed by western blotting with the indicated antibodies. **d** HeLa cells transfected with either ΔNLS-TFEB-GFP or wild-type TFEB-GFP were either starved for 60 min or starved and restimulated with amino acids for 30 min in the presence or absence of 250 nM Torin, and evaluated for TFEB phosphorylation by immunoblotting. **e** HeLa cells stably expressing TFEB-GFP were either starved for amino acids (aa) for 60′, or starved and restimulated with aa for 30′ in the presence or absence of inhibitors of mTOR (Torin; 250 nM), ERK (U0126; 10 µM), and GSK3 (SB415286; 50 µM). Cell extracts were analyzed by western blotting with the indicated antibodies

conditions benefiting from induction or inhibition of TFEB activity, such as neurodegenerative disorders and cancer, respectively.

## Methods

**Cell culture conditions**. Cells were cultured in the following media: HeLa (ATCC) in MEM (Cat# ECB2071L, Euroclone); HEK293T (ATCC) in DMEM high glucose (Cat# ECM0728L, Euroclone); ARPE19 (ATCC) in DMEM-F12 (Cat# 11320–033, Thermo Fisher Scientific). All media were supplemented with 10% inactivated FBS (Cat# ECS0180L, Euroclone), 2 mM glutamine (Cat# ECB3000D, Euroclone), penicillin (100 IU/mL), and streptomycin (100 µg/mL) (Cat# ECB3001D, Euroclone) and maintained at 37 °C and 5% CO$_2$. HeLa cells stably expressing TFEB-GFP were previously described[19].

All cell lines were tested and validated for the absence of mycoplasma.

**Materials**. Reagents used in this study were obtained from the following sources: Antibodies to Phospho-p70 S6 Kinase (Thr389) (1A5) (Cat# 9206) (1:1000), p70 S6 Kinase (Cat# 9202) (1:1000), 4E-BP1 (Cat# 9644) (1:1000), Phospho-4E-BP1 (Ser65) (Cat# 9456) (1:1000), TFEB (Cat# 4240) (1:1000), phospho-GSK3-beta (Ser9) (D3A4) (Cat# 9322) (1:1000), GSK3-beta (27C10) (Cat# 9315) (1:1000), were from Cell Signaling Technology; antibody to GFP (Cat# ab13970) (1:5000) was from Abcam; antibody to GAPDH (6C5) (Cat# sc-32233) (1:20000) was from Santa Cruz; antibody to TFEB-pSer142 (Cat# ABE1971) (1:15000) was from EMD-Millipore; antibodies to ERK1/ERK2 (Cat# MAB1576) (1:1000) and phospho-ERK1 (T202/204)/ERK2 (T185/Y187) (Cat# AF1018) (1:1000) were from R&D Systems; antibodies to TFEB-pS138 (1:15000) were custom generated in

collaboration with Bethyl Laboratories. Both TFEB-pSer142 and TFEB-pS138 antibodies tested in this study showed marginal cross-reactivity with unphosphorylated TFEB (Fig. 4c).

Chemicals: Torin 1 (Cat# 4247) was from Tocris; Leptomycin B (Cat# L2913), Protease Inhibitor Cocktail (Cat# P8340) and puromycin (Cat# P9620) were from Sigma-Aldrich; U0126 (Cat# 9903) was from Cell Signaling; SB415286 (Cat# 1617) was from TOCRIS; PhosSTOP phosphatase inhibitor cocktail tablets (Cat# 04906837001) were from Roche.

**Plasmids**. The plasmid-encoding ΔNLS-TFEB-GFP was a kind gift of Shawn Ferguson (Yale School of Medicine, CT, USA). TFEB S138A-GFP was a kind gift of Chonglin Yang (Chinese Academy of Sciences, China). Human full-length TFEB-GFP wt was previously described[3]. Human TFEB S138A-GFP, TFEB S142A-GFP, TFEB S211A-GFP, TFEB S142/211A-GFP, TFEB M144A-GFP, TFEB L147A-GFP, and TFEB I149A-GFP were generated using QuikChange II-XL Site-Directed Mutagenesis Kit (Cat# 200522, Agilent Technologies).

**Cell treatments**. For experiments involving amino acid starvation, cell culture plates were rinsed twice with PBS and incubated in amino acid-free RPMI (Cat# R9010-01, USBiological) supplemented with 10% dialyzed FBS for 60 min. Where indicated, cells were restimulated for 30 min with 1× water-solubilized mix of essential (Cat#11130036, Thermo Fisher Scientific) and non-essential (Cat# 11140035, Thermo Fisher Scientific) amino acids resuspended in amino acid-free RPMI. Where reported, cells were incubated with 5 nM Leptomycin B or 250 nM Torin 1 during amino acid restimulation. For evaluation of TFEB nuclear phosphorylation, Leptomycin B (5 nM) was also used during starvation as a pretreatment to maximize TFEB nuclear retention.

For siRNA-based experiments, cells were transfected using Lipofectamine® RNAiMAX Transfection Reagent (#13778, Invitrogen) with the indicated siRNAs and analyzed after 72 h.

The following siRNA were used:
siCRM1: #1 CUAUGAGGAAUGUCGCAGA;
#2 GGAUAUCAACUUAUUAGAU;
#3 CCAAUAUUCGACUUGCGUA
Control non-targeting siRNA Pool (D-001810-10-05) were from Dharmacon.

**High-content analysis**. HeLa TFEB-GFP cells were seeded in 96-well plates and incubated for 24 h. After incubation cells were treated as described above, rinsed with PBS once, fixed for 10 min with 4% paraformaldehyde and stained with DAPI. For the acquisition of the images, at least ten image fields were acquired per well of the 96-well plate by using confocal automated microscopy (Opera high content system; Perkin-Elmer). A dedicated script was developed to perform the analysis of TFEB localization on the different images (Harmony and Acapella software; Perkin-Elmer). The script calculates the ratio value resulting from the average intensity of nuclear TFEB-GFP fluorescence divided by the average of the cytosolic intensity of TFEB-GFP fluorescence. *p* values were calculated on the basis of mean values from independent wells.

**Cell lysis and western blotting**. Cells were rinsed once with PBS and lysed in ice-cold lysis buffer (250 mM NaCl, 1% Triton, 25 mM Hepes pH 7.4) supplemented with protease and phosphatase inhibitors. Total lysates were passed ten times through a 25-gauge needle with syringe, kept at 4 °C for 10 min and then cleared by centrifugation in a microcentrifuge ($21,000 \times g$ at 4 °C for 10 min). Protein concentration was measured by Bradford assay. Denatured samples were resolved by SDS-polyacrylamide gel electrophoresis on 4–12% Bis-Tris gradient gels (Cat# NP0323PK2 NuPage, Thermo Fischer Scientific), transferred to PVDF membranes and analyzed by immunoblotting with the indicated primary antibodies.

Where reported, cells were transfected in 10 cm dishes using Fugene 6 (Promega) with the following plasmids and quantities: 1 µg TFEB WT-GFP, 1 µg TFEB S138A-GFP, 2 µg TFEB S142A-GFP, 2.5 µg TFEB S211A-GFP, 1 µg TFEB S142/211A-GFP, 150 ng TFEB ΔNLS-GFP, 1.5 µg TFEB-M144A-GFP, 1.5 µg TFEB-L147A-GFP, and 1.5 µg TFEB-I149A-GFP. The total amount of transfected plasmid DNA in each transfection was normalized to 3 µg using an empty plasmid.

Uncropped scans of the most relevant blots are included in Supplementary Fig. 6.

**Confocal microscopy**. Immunofluorescence experiments were performed as previously described[39]. Briefly, cells were grown on 8-well Lab-Tek II-Chamber Slides, treated as indicated, and fixed with 4% parafolmaldehyde (PFA) for 10 min at RT. For endogenous TFEB staining, cells were permeabilized with 0.1% Triton X-100 for 5 min, followed by blocking with 3% bovine serum albumin in PBS + 0.02% saponin for 1 h at RT. Immunostainings were performed upon dilution of primary antibodies in blocking solution and overnight incubation at 4 °C, followed by three washes and secondary antibody incubation in blocking solution for 1 h at RT. After additional three washes, coverslips were finally mounted in VECTASHIELD® mounting medium with DAPI and analyzed using LSM 700 or LSM 800 with a Plan-Apochromat ×63/1.4 NA M27 oil immersion objective using immersion oil (#518F, Carl Zeiss) at room temperature. The microscopes were operated on the ZEN 2013 software platform (Carl Zeiss).

**RNA extraction and real-time PCR**. Total RNA was extracted from cellular lysates using the RNeasy Mini kit (Cat# 74104, Qiagen) according to the manufacturer's instructions. cDNA was synthesized by reverse transcription of total RNA (500 ng per sample) using QuantiTect Reverse Transcription kit (Cat# 205311, Qiagen).

The generated cDNA was diluted six-fold and used as a template for real-time quantitative PCR, which was performed with the LightCycler 480 SYBR Green I mix (Cat# 04 887 352 001, Roche) using the Light Cycler 96 detection system (Roche). Melting curve analyses were performed to verify the amplification specificity. The housekeeping gene HPRT was used as an internal control to normalize the variability in expression levels. Relative quantification of gene expression was performed according to the 2 $(-\Delta\Delta CT)$ method.

The forward and reverse primers for HPRT were 5′-tggcgtcgtgattagtgatg-3′ and 5′-aacacccttttccaaatcctca-3′ and for CRM1 were 5′-AGCAAAGAATGGCTCAAGAAGT-3′ and 5′-TATTCCTTCGCACTGGTTCCT-3′

**FRAP/FLIP and live cell imaging experiments**. For fluorescence recovery after photobleaching (FRAP) experiments, HeLa cells stably expressing TFEB-GFP seeded on 35 mm glass bottom dishes (MatTek) were starved for 60′ in HBSS + 10 mM Hepes (starvation medium), and then medium was replaced with either starvation medium or with phenol red-free complete DMEM in the presence or absence of either 250 nM Torin or 5 nM Leptomycin B for 30′. Cells were imaged using a LSM 880 + Airyscan systems (Carl Zeiss) with a 488 nm laser through a ×63 oil immersion objective at 37 °C and 5% CO₂. A region of interest comprising the whole cytosol was designed and movie acquisition was started. After ten frames, cells were photobleached with three consecutive 488 nm pulses, each spaced out by three acquisition frames, and then imaged for 15 min. For fluorescence loss in

photobleaching (FLIP) experiments, cells were photobleached every frame until the end of the imaging.

For FRAP experiments involving mutant TFEB, cells were directly transfected on 8-well Nunc™ Lab-Tek™ Chambered Coverglass (Thermo Fisher) using a Fugene 6 transfection mixture containing each of the following plasmids: 200 ng TFEB WT-GFP, 200 ng TFEB S138A-GFP, 400 ng TFEB S142A-GFP, 500 ng TFEB 211A-GFP, 200 ng TFEB M144A-GFP, 200 ng TFEB L147A-GFP, and 200 ng TFEB I149A-GFP, which were normalized to 1 µg total DNA using an empty plasmid. One-tenth of each transfection mixture was used for reverse transfection and cells were analyzed as above no longer than 24 h after transfection. FRAP and FLIP experiments were then analyzed using the ImageJ software for the calculation of nuclear and cytosolic intensity. The decay of nuclear intensity in FRAP and FLIP experiments was calculated as the nuclear intensity divided for the total cell fluorescence intensity (cytosolic + nuclear intensity) in each frame, starting from the first frame after photobleaching to the end of the acquisition.

**Data availability**. The data that support the findings of this study are available from the corresponding author on reasonable request.

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

## Acknowledgements

We thank Ya-Cheng Liao for technical help. This work was supported by grants from the Italian Telethon Foundation (TGM16CB6); MIUR FIRB RBAP11Z3YA (A.B.); European Research Council Advanced Investigator no. 694282 (LYSOSOMICS) (A.B.); U.S. National Institutes of Health (R01-NS078072) (A.B.); the Huffington foundation (A.B.); and the Associazione Italiana per la Ricerca sul Cancro (A.I.R.C.) (A.B.) (IG 2015 Id 17639). G.N. was supported by the European Union's Horizon 2020 research and innovation program under the Marie Sklodowska-Curie Grant Agreement No 661271.

## Author contributions

G.N. and A.B. conceived the project; G.N., J.L.S., and A.B. designed the experiments; G. N., A.E., H.C., M.M., V.B., C.D.M., and J.M. performed the experiments; D.L.M., J.L.S., and A.B. provided resources and supervision; G.N. and A.B. wrote the manuscript.

## Additional information

**Competing interests:** The authors declare no competing interests.

