## [Peer Review File · Nature Communications]

Reviewers' comments:

Reviewer #1 (Remarks to the Author):

The paper by Napolitano et al identifies a mechanism that governs the nuclear-cytoplasmic distribution of the master transcription factor TFEB and may help coordinate nutrient status with the induction of TFEB-dependent catabolic programs. The authors demonstrate that TFEB harbors in its N-terminal region a nuclear export signal that is recognized by the CRIM1 nuclear export factor. Upon phosphorylation of this region by the mTOR kinase, export of TFEB from the nucleus is accelerated. Conversely, mutating key residues in the NES or nearby phosphor-sites results in nuclear retention of TFEB, due to greatly decreased export rates.

The manuscript proposes an elegant model by which TFEB cycles between the nucleus and the cytoplasm via the combined action of import and export factors. At different points along this cycle TFEB would be intercepted by its regulatory kinases and phosphatases in a nutrient-regulated manner. The manuscript will surely be of interest to the field of metabolic regulation, and it is supported by high-quality data.

A few points for clarification are noted below.

- 1- In Fig. 2D, in addition to plots showing decrease of nuclear GFP-TFEB signal over time, increase of cytoplasmic signal should also be plotted.
- 2- In Fig. 3E, the cytoplasmic signal should be plotted as well.
- 3- In Fig. 5A, a residual signal in the p138 and p142 antibodies is present in the Torin-treated samples. Are these due to cross-reactivity of the phosphor antibodies toward total TFEB? If so, this should be mentioned in the text, particularly if these reagents are unpublished.
- 4- In Fig. 5C, S138 and S142 are phosphorylated on M144A (nuclear-retained) TFEB mutant with similar efficiency to the wild-type protein, despite its retention in the nucleus. Presumably, this is due to the fact that the M144A mutant TFEB is still exported from the nucleus, albeit at decreased rate (Fig. 3F). Are the kinetics of phosphorylation identical between WT and M144A TFEB? The authors should perform a time-course of amino acid restimulation (e.g. 1, 5, 15, 45, 120 min) and compare the phosphorylation time curves of the two isoforms.
- 5- The observation that the NLS-deleted TFEB is less phosphorylated than WT suggests that TFEB may be phosphorylated by mTOR on its way out of the nucleus, not on its way in. The phosphorylation time-course suggested above should help clarify this point.
- 6- It is recommended that the authors provide a model that summarizes their findings at the end of Fig. 5

Minor points:

- 1- in Fig. 1C plot, the line connecting the dots gives the impression that this is a time course, which it is not. If less than 10 experimental points (i.e. cells) per conditions were counted, these should be shown individually. Same for 2A.
- 1- in Fig. 3D, it seems odd that the % of cells with nuclear TFEB in the fed state is exactly

zero. Even if it is, the graph should be re-plotted in a way that makes this data group visible in order not to confuse the reader.

Reviewer #2 (Remarks to the Author):

In this manuscript, authors reported a new mechanism that nutrient (amino acid, aa) controls TFEB shuttling between the cytosol and the nucleus. In the presence of aa, TFEB is phosphorylated by mTOR which plays a crucial role in determining TFEB cytosolic localization. Upon starvation stress, inhibition of mTOR and concomitant activation of the phosphatase calcineurin by TRPML1-mediated lysosomal calcium release induces TFEB dephosphorylation, and which leads to a nuclear localization of TFEB. Moreover, this manuscript reveals that nutrient promotes cytosolic re-localization of nuclear TFEB via CRM1-dependent nuclear export.

As part of the mechanisms, authors demonstrated the possess of a nuclear export signal (NES) localized in the N-terminal portion of TFEB, whose integrity is absolutely required for TFEB nuclear export. And they found that nutrient- and mTOR-dependent phosphorylation of S142 and S138, which are localized in proximity of the NES, is necessary to induce TFEB nuclear export and lose its activity. However, similar results about the N-terminal NES of TFEB had reported in the paper "The Transcription Factor TFEB Links mTORC1 Signaling to Transcriptional Control of Lysosome Homeostasis" Sci Signal (2012, Jun 12), and in this regard, this largely weakened the novelty of the study.

In summary, authors provided the data to support their conclusions and showed us a new mechanism by which nutrient availability controls TFEB localization and activity. Overall, the manuscript is well written, the experiments are conducted in a logical fashion, and the figures are well plotted and clear. However, there are several concerns which need to be addressed before it can be accepted for publication.

1. Authors performed their experiments mostly in HeLa cells. Does the mechanism that nutrient controls TFEB localization commonly occurs in most other cell lines?
2. How did the authors originally find that CRM1 has a transport effect on TFEB? Authors should explain it clearly.
3. For the results of figure 1-F, the knock down efficiency of CRM1 should be shown.
4. TFEB predominantly localized in cytosol in fed cells, and why the Nuc/Cyt ratio of TFEB is about 1 in the figure 1-C?
5. Does the CRM1 directly bind to TFEB to help its nuclear export? Need experiments to delineate it.
6. Do the other nutrients (eg. glucose, FBS etc) have a similar impact on the localization of TFEB in addition to amino acids?
7. What is the mechanism of TFEB entering the nucleus after its dephosphorylation upon starvation stress? This is also important for us to have a better understanding of the regulation of the shuttle for TFEB.
8. TFEB has been fully phosphorylated under the nutrition conditions, how can the phosphorylated-TFEB enter into nucleus?

Reviewer #3 (Remarks to the Author):

Napolitano et al. reported in this paper that mTOR-dependent Ser138 and 142 phosphorylation of TFEB within the nucleus induced the nuclear export of the phosphorylated TFEB in a manner dependent of XPO1/CRM1. While It has been well documented that upon amino acid starvation or mTORC1 inhibition, the dephosphorylated TFEB translocates into the nucleus as an active form, it has remained unclear how the dephosphorylated active nuclear TFEB is phosphorylated and exported from the nucleus in response to nutrients replenishment and mTORC1 reactivation. The authors demonstrated that Ser138 of TFEB was phosphorylated in a manner dependent of Ser142 phosphorylation by mTOR, which facilitates the nuclear export of TFEB in a manner dependent on the CRM1. The authors identified a putative CRM1-NES site on the TFEB, and the mutations of the putative CRM1-binding site in fact blocked TFEB's nuclear export in response to the replenishment of amino acids. Interestingly, both Ser138 and Ser142 of TFEB, the sites phosphorylated by mTOR and important for TFEB's nuclear export, overlapped with the CRM1 NES, suggesting that the phosphorylation of Ser 138 and 142 may positively contribute to the regulation of TFEB binding to CRM1.

The experiments were nicely designed and the data demonstrated in this manuscript were clean and convincing, although XPO1/CRM1-dependent nuclear export of TFEB has just been reported by the other group. The study will be more strengthened if the authors put additional information listed below.

Comments:

1. It would be important to confirm the effect of CRM1 inhibition on the accumulation of endogenous nuclear TFEBs.
2. It would be important to demonstrate biochemical evidence for nutrients- and mTOR activity-dependent TFEB-CRM1 interaction and the roles of Ser138/142 phosphorylation in the interaction between TFEB and CRM1.
3. Fig.4 suggested an important role of the hierarchical TFEB phosphorylation of Ser142 followed by Ser138 by mTOR kinase for its nuclear export. Do the phospho-mimetic mutations of these phosphorylation sites cause a cytoplasmic localization of TFEB under starvation conditions? Can the S142A/S138D TFEB mutant be exported from the nucleus under starvation conditions?
4. Although the data in Fig. 5 suggested that the mTOR kinase phosphorylated the M144A TFEB in the nucleus in response to amino acid stimulation, it remains unclear if and how the mTORC1 or atypical mTOR complex phosphorylates TFEB in the nucleus. Upon amino acids replenishment, does the lysosome-mediated mTORC1 activation still play an important role in phosphorylating nuclear TFEBs? What is the effect of active Rag heterodimer on the phosphorylation of the M144A TFEB under amino acid starvation conditions? Can the Raptor-15aaRheb, which is able to stimulate the lysosome-mediated mTORC1 activation in a

manner independent of the Ragulator-Rag system or amino acid availability, enhance the phosphorylation of the M144A TFEB under amino acid starvation conditions?

Reviewers' comments:

Reviewer #1 (Remarks to the Author):

The paper by Napolitano et al identifies a mechanism that governs the nuclear-cytoplasmic distribution of the master transcription factor TFEB and may help coordinate nutrient status with the induction of TFEB-dependent catabolic programs. The authors demonstrate that TFEB harbors in its N-terminal region a nuclear export signal that is recognized by the CRIM1 nuclear export factor. Upon phosphorylation of this region by the mTOR kinase, export of TFEB from the nucleus is accelerated. Conversely, mutating key residues in the NES or nearby phosphor-sites results in nuclear retention of TFEB, due to greatly decreased export rates.

The manuscript proposes an elegant model by which TFEB cycles between the nucleus and the cytoplasm via the combined action of import and export factors. At different points along this cycle TFEB would be intercepted by its regulatory kinases and phosphatases in a nutrient-regulated manner. The manuscript will surely be of interest to the field of metabolic regulation, and it is supported by high-quality data.

We thank the reviewer for positively evaluating our work.

A few points for clarification are noted below.

1- In Fig. 2D, in addition to plots showing decrease of nuclear GFP-TFEB signal over time, increase of cytoplasmic signal should also be plotted.

The plots of the cytosolic signal for Fig. 2D are now shown in the new Supplementary Fig. 2.

2- In Fig. 3E, the cytoplasmic signal should be plotted as well.

The plots of the cytosolic signal for Fig. 3E are now shown in the new Supplementary Fig. 3.

3- In Fig. 5A, a residual signal in the p138 and p142 antibodies is present in the Torin-treated samples. Are these due to cross-reactivity of the phosphor antibodies toward total TFEB? If so, this should be mentioned in the text, particularly if these reagents are unpublished.

Although phospho-antibodies against p138 (unpublished) and S142 (previously published) are highly sensitive and specific, they partially recognize the unphosphorylated form of TFEB, as we still observe a faint signal when we use the same antibodies using the S142A and S138A mutants (Fig. 4C). This is now mentioned in the Materials and Methods.

4- In Fig. 5C, S138 and S142 are phosphorylated on M144A (nuclear-retained) TFEB mutant with similar efficiency to the wild-type protein, despite its retention in the nucleus. Presumably, this is due to the fact that the M144A mutant TFEB is still exported from the nucleus, albeit at decreased rate (Fig. 3F). Are the kinetics of phosphorylation identical between WT and M144A TFEB? The authors should perform a time-course of amino acid restimulation (e.g. 1, 5, 15, 45, 120 min) and compare the phosphorylation time curves of the two isoforms.

We would like to thank the reviewer for this helpful suggestion. We performed the time-course experiment suggested by the reviewer. The results showed that the phosphorylation of the M144A TFEB nuclear export mutant is highly efficient, and even enhanced, at early

time points upon re-feeding compared to WT TFEB (new Supplementary Fig. 4A). This experiment further suggests that TFEB phosphorylation may occur in the nuclear compartment.

5- The observation that the NLS-deleted TFEB is less phosphorylated than WT suggests that TFEB may be phosphorylated by mTOR on its way out of the nucleus, not on its way in. The phosphorylation time-course suggested above should help clarify this point.

We completely agree with the reviewer. Our data suggest that mTOR phosphorylates TFEB on its way out of the nucleus, presumably before nuclear export occurs. This hypothesis is supported by multiple pieces of evidence: firstly, TFEB is efficiently re-phosphorylated upon re-feeding in leptomycin-treated cells, in which TFEB shows a predominantly nuclear localization as a result of impaired nuclear export (Fig. 5A-5B); secondly, the M144A TFEB mutant shows efficient re-phosphorylation upon re-feeding (Fig 5C) despite its nuclear export is dramatically impaired (Fig. 3C-3F); thirdly, the constitutively nuclear S138A TFEB mutant also shows efficient S142 phosphorylation (Fig. 4C); finally, the new time course experiment suggested by this reviewer further supports the hypothesis that mTOR-mediated TFEB phosphorylation may occur in the nucleus (new Supplementary Fig. 4A). However, we wish to clarify that more data are needed in order to characterize the identity of this nuclear pool of mTOR and to understand how it is regulated. We believe that this is out of the scope of the present manuscript.

This point is now extensively addressed in the discussion.

6- It is recommended that the authors provide a model that summarizes their findings at the end of Fig. 5

We now provide a model in Supplementary Fig. 5.

Minor points:

1- in Fig. 1C plot, the line connecting the dots gives the impression that this is a time course, which it is not. If less than 10 experimental points (i.e. cells) per conditions were counted, these should be shown individually. Same for 2A.

We apologize for the lack of clarity on this point. The experiments in Figure 1C and 2A represent a high content analysis of cells plated in 96 well plates, treated as indicated, and analyzed with the OPERA system, which allows the automated acquisition and analysis of hundreds of cells per well (as described in the Methods section). Please note that, in each experiment, each treatment is performed in triplicate. Therefore, each dot shown in Fig 1C and 2A represents the average nucleo/cytosolic ratio of TFEB in almost one thousand cells. The graph is representative of three different experiments done in the same conditions. This is now better explained in the figure legend.

1- in Fig. 3D, it seems odd that the % of cells with nuclear TFEB in the fed state is exactly zero. Even if it is, the graph should be re-plotted in a way that makes this data group visible in order not to confuse the reader.

Re-feeding strongly induces TFEB cytosolic re-localization in almost 100% of the cells, which makes the % of cells with nuclear TFEB in the re-feeding state close to zero. However, we agree with the reviewer that the graph was confusing and analyzed multiple other fields. A new clearer graph with such quantification is now shown in Fig. 3D.

Reviewer #2 (Remarks to the Author):

In this manuscript, authors reported a new mechanism that nutrient (amino acid, aa) controls TFEB shuttling between the cytosol and the nucleus. In the presence of aa, TFEB is phosphorylated by mTOR which plays a crucial role in determining TFEB cytosolic localization. Upon starvation stress, inhibition of mTOR and concomitant activation of the phosphatase calcineurin by TRPML1-mediated lysosomal calcium release induces TFEB dephosphorylation, and which leads to a nuclear localization of TFEB. Moreover, this manuscript reveals that nutrient promotes cytosolic re-localization of nuclear TFEB via CRM1-dependent nuclear export.

As part of the mechanisms, authors demonstrated the possess of a nuclear export signal (NES) localized in the N-terminal portion of TFEB, whose integrity is absolutely required for TFEB nuclear export. And they found that nutrient- and mTOR-dependent phosphorylation of S142 and S138, which are localized in proximity of the NES, is necessary to induce TFEB nuclear export and lose its activity. However, similar results about the N-terminal NES of TFEB had reported in the paper "The Transcription Factor TFEB Links mTORC1 Signaling to Transcriptional Control of Lysosome Homeostasis" Sci Signal (2012, Jun 12), and in this regard, this largely weakened the novelty of the study.

In summary, authors provided the data to support their conclusions and showed us a new mechanism by which nutrient availability controls TFEB localization and activity. Overall, the manuscript is well written, the experiments are conducted in a logical fashion, and the figures are well plotted and clear. However, there are several concerns which need to be addressed before it can be accepted for publication.

We thank the reviewer for positively evaluating our work.

However, we would like to underline that the work by Roczniak-Ferguson et al. (Sci signal 2012) does not weaken in any way the novelty of our work. The paper by Roczniak-Ferguson et al. did not contain any data on TFEB Nuclear Export Signal and in fact Nuclear Export was not even mentioned in that paper. Roczniak-Ferguson et al. characterized TFEB Nuclear Localization Signal (NLS), which is required for TFEB nuclear import (not the export!). In this manuscript, we identified a Nuclear Export Signal (NES), proximal to amino acids S142 and S138, required for TFEB nuclear export. This is the first evidence of a NES-mediated nutrient-dependent mechanism that controls TFEB nuclear export.

1. Authors performed their experiments mostly in HeLa cells. Does the mechanism that nutrient controls TFEB localization commonly occurs in most other cell lines?

The nutrient-dependent control of TFEB subcellular localization has already been demonstrated by several groups in multiple cell lines, such as HeLa, ARPE19 and HEK293T, as well as in vivo in tissues from normally fed and starved mice. Following the reviewer's suggestion, we have expanded our analysis of TFEB nuclear export to a total of 3 different cell lines: HeLa, HEK293T and ARPE19 (Fig. 1 and new Supplementary Figure 1).

2. How did the authors originally find that CRM1 has a transport effect on TFEB? Authors should explain it clearly.

The dynamic analysis of nutrient-dependent TFEB redistribution from the nucleus to the cytosol (Fig. 1A) suggested that TFEB underwent active nuclear export. In addition, a previous interactome analysis of CRM1 binding proteins showed that TFEB is a strong

CRM1-interacting protein (Kirli et al 2015, also see point 5). These data prompted us to test the effect of CRM1 on TFEB nuclear export. This is now better explained in the text.

3. For the results of figure 1-F, the knock down efficiency of CRM1 should be shown. This is now shown in new Supplementary Fig. 1A.

4. TFEB predominantly localized in cytosol in fed cells, and why the Nuc/Cyt ratio of TFEB is about 1 in the figure 1-C?

The Nuc/Cyt ratio shown in Figure 1C was calculated using the OPERA system, as described in previous publications (Settembre et al. EMBO J, 2012; Medina et al. Nat Cell Biol, 2015) and explained in the Methods section of the present manuscript. The Nuc/Cyt ratio is calculated from a population of cells by using the software Harmony from Perkin Elmer (Settembre et al. EMBO J, 2012; Medina et al. Nat Cell Biol, 2015). The script calculates the ratio value resulting from the average intensity of nuclear TFEB–GFP fluorescence divided by the average of the cytosolic intensity of TFEB–GFP fluorescence in the cell population. Depending on the cell line this ratio is between 0.8 and 1 in normally fed cells and between 1.4 and 1.8 in starved cells (Settembre et al. EMBO J, 2012; Medina et al. Nat Cell Biol, 2015). The reason why the values in normally fed cells are close to 1 is likely related to the fact that the size of the nucleus is much smaller than the cytoplasm and this may increase the background.

5. Does the CRM1 directly bind to TFEB to help its nuclear export? Need experiments to delineate it.

We agree with the reviewer that this would be an interesting experiment to perform. However, the CRM1-binding assay is very complicated as it requires in vitro production of three different recombinant proteins at very high purity, followed by affinity chromatography (please see Kirli et al 2015 for details). We have no experience with this assay, therefore, several months ago we decided to start a collaboration with the group of an expert in nucleocytoplasmic shuttling, to try to answer this issue. However, after several months of work, they are still trying to get the assay properly done due to technical issues. Therefore, we believe that this issue cannot be answered within a reasonable time. Please note, however, that TFEB has already been shown to bind to CRM1 in a deep proteomic analysis aimed at identifying novel CRM1 binders (Kirli et al eLife, 2015). Strikingly, TFEB ranked as the second best CRM1 binder among hundreds of different binders! This work, together with our data that TFEB nuclear export is Leptomycin- and CRM1-dependent, provides very strong evidence for a role of CRM1 in directly modulating TFEB nuclear export.

6. Do the other nutrients (eg. glucose, FBS etc) have a similar impact on the localization of TFEB in addition to amino acids?

In our paper we have studied the effects of amino acid starvation simply because this is the best described physiological condition that is known to inactivate mTOR and, in particular, to inhibit mTOR-mediated TFEB phosphorylation. In our hands neither FBS nor glucose starvation have significant effects on TFEB subcellular localization (see Figure attached below, similar results were obtained with up to 6h starvation). Nevertheless, we cannot rule out the possibility that under certain conditions or in different cell lines either FBS or glucose may affect TFEB localization. However, considering also that the modulation of mTOR

activity by glucose is controversial and poorly understood, we have decided to focus our manuscript on amino acid-mediated modulation of TFEB.

GFP-TFEB expressing HeLa cells were starved of either amino acids (AA), FBS or glucose for 60 minutes and analyzed by confocal microscopy.

7. What is the mechanism of TFEB entering the nucleus after its dephosphorylation upon starvation stress? This is also important for us to have a better understanding of the regulation of the shuttle for TFEB.

In the presence of nutrients, TFEB phosphorylation on S211 has been shown to serve as a binding site for the chaperone 14-3-3, which allows TFEB cytosolic retention by masking a nuclear localization signal (NLS) localized at the C-terminus of the protein (Martina et al. 2012; Roczniak-Ferguson et al. 2012). Upon starvation, activation of the phosphatase Calcineurin induces TFEB de-phosphorylation (Medina et al. 2015) and subsequent 14-3-3 dissociation, which un masks the NLS and allows nuclear translocation of TFEB (Martina et al. 2012; Roczniak-Ferguson et al. 2012). The present manuscript adds a new layer of complexity by identifying TFEB nuclear export as a main limiting step in the modulation of TFEB subcellular localization and by revealing a previously uncharacterized role for S142 and S138 as crucial residues required for NES-mediated TFEB nuclear export.

8. TFEB has been fully phosphorylated under the nutrition conditions, how can the phosphorylated-TFEB enter into nucleus?

We show that TFEB continuously shuttles between the cytosol and the nucleus both in the presence and absence of nutrients (Figure 2), although at different rates (Figure 3G and 3H), as previously shown for many other transcription factors that are subject to nucleo-cytoplasmic shuttling. We propose that, in the presence of nutrients, basal levels of cytosolic phosphatase activity induce constant TFEB de-phosphorylation and nuclear translocation.

Reviewer #3 (Remarks to the Author):

Napolitano et al. reported in this paper that mTOR-dependent Ser138 and 142 phosphorylation of TFEB within the nucleus induced the nuclear export of the phosphorylated TFEB in a manner dependent of XPO1/CRM1. While It has been well documented that upon amino acid starvation or mTORC1 inhibition, the dephosphorylated TFEB translocates into the nucleus as an active form, it has remained unclear how the dephosphorylated active nuclear TFEB is phosphorylated and exported from the nucleus in response to nutrients replenishment and mTORC1 reactivation. The authors demonstrated that Ser138 of TFEB was phosphorylated in a manner dependent of Ser142 phosphorylation by mTOR, which facilitates the nuclear export of TFEB in a manner dependent on the CRM1. The authors identified a putative CRM1-NES site on the TFEB, and the mutations of the

putative CRM1-binding site in fact blocked TFEB's nuclear export in response to the replenishment of amino acids. Interestingly, both Ser138 and Ser142 of TFEB, the sites phosphorylated by mTOR and important for TFEB's nuclear export, overlapped with the CRM1 NES, suggesting that the phosphorylation of Ser 138 and 142 may positively contribute to the regulation of TFEB binding to CRM1.

The experiments were nicely designed and the data demonstrated in this manuscript were clean and convincing, although XPO1/CRM1-dependent nuclear export of TFEB has just been reported by the other group. The study will be more strengthened if the authors put additional information listed below.

We thank the reviewer for positively evaluating our work.

Comments:

1. It would be important to confirm the effect of CRM1 inhibition on the accumulation of endogenous nuclear TFEBs.

We expanded our analysis of TFEB nuclear export in other cell lines and corroborated our findings that TFEB undergoes CRM1-dependent nuclear export in HEK293T and ARPE-19 cell lines (Figure 1D and Supplementary Fig. 1B-1C).

2. It would be important to demonstrate biochemical evidence for nutrients- and mTOR activity-dependent TFEB-CRM1 interaction and the roles of Ser138/142 phosphorylation in the interaction between TFEB and CRM1.

As discussed in point 5 of reviewer 2, we agree with the reviewers that this would be an interesting experiment to perform. However, the CRM1-binding assay is very complicated as it requires in vitro production of three different recombinant proteins at very high purity, followed by affinity chromatography (please see Kirli et al 2015 for details). We have no experience with this assay, therefore, several months ago we decided to start a collaboration with the group of an expert in nucleo-cytoplasmic shuttling, to try to answer this issue. However, after several months of work, they are still trying to get the assay properly done due to technical issues. Therefore, we believe that this issue cannot be answered within a reasonable time. Please note, however, that TFEB has already been shown to bind to CRM1 in a deep proteomic analysis aimed at identifying novel CRM1 binders (Kirli et al eLife, 2015). Strikingly, TFEB ranked as the second best CRM1 binder among hundreds of different binders! This work, together with our data that TFEB nuclear export is Leptomycin- and CRM1-dependent, provides very strong evidence for a role of CRM1 in directly modulating TFEB nuclear export. In addition, the impaired export kinetics of TFEB phosphorylation mutants further supports a role of TFEB phosphorylation in CRM1-mediated nuclear export. However, it remains to be demonstrated whether or not the binding of TFEB to CRM1 is directly dependent on TFEB phosphorylation. Unfortunately, for the above outlined reasons, addressing this point appears to be very challenging.

3. Fig.4 suggested an important role of the hierarchical TFEB phosphorylation of Ser142 followed by Ser138 by mTOR kinase for its nuclear export. Do the phospho-mimetic mutations of these phosphorylation sites cause a cytoplasmic localization of TFEB under starvation conditions? Can the S142A/S138D TFEB mutant be exported from the nucleus under starvation conditions?

We thank the reviewer for this suggestion. We generated an S142D/S138D phospho-mimetic TFEB mutant and an S142A/S138D TFEB mutant, as indicated by the reviewer. In addition, in order to avoid that de-phosphorylation of other serine residues (e.g. S211) may affect TFEB localization even in the context of the S142D/S138D mutant, we decided to generate and evaluate the subcellular localization of a triple S142D/S138D/S211D phospho-mimetic TFEB mutant. As shown in the Figure below, however, none of the TFEB mutants was capable of maintaining a cytosolic localization during starvation. Conversely and contrary to our expectations, all the mutants appeared to be partially nuclear even during re-feeding. Although we cannot exclude that additional serine residues still affect the subcellular localization of the TFEB mutants, these data suggest that TFEB phospho-mimetic mutations do not recapitulate the physiological role of TFEB phosphorylation (phospho-mimetic mutants sometimes do not mimic the physiological effect of phosphorylation, as often observed for other proteins). For this reason, we have decided to exclude these data from the manuscript.

HeLa cells were transiently transfected with the indicated TFEB mutants or with WT TFEB, subjected to starvation (-aa) and re-feeding (+aa) and analysed by confocal microscopy.

4. Although the data in Fig. 5 suggested that the mTOR kinase phosphorylated the M144A TFEB in the nucleus in response to amino acid stimulation, it remains unclear if and how the mTORC1 or atypical mTOR complex phosphorylates TFEB in the nucleus. Upon amino acids replenishment, does the lysosome-mediated mTORC1 activation still play an important role in phosphorylating nuclear TFEBs? What is the effect of active Rag heterodimer on the phosphorylation of the M144A TFEB under amino acid starvation conditions? Can the Raptor-15aaRheb, which is able to stimulate the lysosome-mediated mTORC1 activation in a manner independent of the Ragulator-Rag system or amino acid availability, enhance the phosphorylation of the M144A TFEB under amino acid starvation conditions?

We agree with the reviewer that these are important issues and, indeed, understanding how lysosomal mTOR contributes to the modulation of TFEB is currently the subject of intensive study in our lab. In the present manuscript, however, we have focused on the nuclear events responsible for the export of TFEB from the nucleus, which is likely due to a different nuclear pool of mTOR. Therefore, we believe that addressing how the lysosomal pool of mTOR functionally cooperates with a putative nuclear pool of mTOR in the regulation of TFEB would be out of the scope of this manuscript, as it would require extensive work and time. In order

to corroborate this point, we would like to share with the reviewer part of our unpublished data showing that the modulation of TFEB by lysosomal mTOR might not be as straightforward as previously thought. As shown in the Figure below and in striking contrast to our expectations, depletion of the well-characterized mTORC1 and mTORC2 components Raptor and Rictor, respectively, has no effect on TFEB phosphorylation, despite a marked impairment in the canonical mTORC1/mTORC2 canonical substrates. As expected, however, TFEB phosphorylation was completely impaired upon mTOR depletion. These data, together with other preliminary data generated in our lab, will eventually lead to a re-visitation of the role of mTORC1, and of mTOR in general, in the regulation of TFEB. As mentioned before, however, we hope the reviewer agrees with us that such characterization requires extensive investigation and is out of the scope of the present manuscript.

HeLa cells were transfected with siRNA targeting Raptor, Rictor, mTOR or with scramble siRNA. 72h after transfection cells were analysed by either immunoblotting (left) or by confocal microscopy (right).

REVIEWERS' COMMENTS:

Reviewer #1 (Remarks to the Author):

The Authors have satisfactorily addressed the concerns raised during the initial submission. Speedy acceptance of the manuscript is recommended.

Reviewer #2 (Remarks to the Author):

Authors had answered most of the questions and made us have a better understanding of their work. However, there still remained a few concerns. Firstly, authors claimed that the nuc/cyt values in normally fed cells are close to 1 is likely due to the fact that size of the nucleus is much smaller than that of the cytoplasm; however, we can clearly see that the size of the nucleus is even larger than the cytoplasm from the images shown in Fig.1. Please explain that again. Is there any other potential mechanism that regulates the localization of TFEB? Secondly, as the reviewer 3 mentioned that XPO1/CRM1-dependent nuclear export of TFEB has been reported in Cell Rep. 2018 May 15, therefore to know the TFEB-CRM1 direct interaction is indispensable for this manuscript. Authors replied in their rebuttal that they are unable to perform this biochemical experiment and that answer is unacceptable.

Reviewer #3 (Remarks to the Author):

The authors' responses are acceptable and the revised paper has been improved. This reviewer appreciated the authors for sharing interesting unpublished observations.

Reviewer #1 (Remarks to the Author):

The Authors have satisfactorily addressed the concerns raised during the initial submission. Speedy acceptance of the manuscript is recommended.

We would like to thank the reviewer for positively evaluating our work.

Reviewer #2 (Remarks to the Author):

Authors had answered most of the questions and made us have a better understanding of their work. However, there still remained a few concerns. Firstly, authors claimed that the nuc/cyt values in normally fed cells are close to 1 is likely due to the fact that size of the nucleus is much smaller than that of the cytoplasm; however, we can clearly see that the size of the nucleus is even larger than the cytoplasm from the images shown in Fig.1. Please explain that again.

We apologise for not being exhaustive enough in our explanation. As indicated in the paper, we use the Perkin-Elmer Opera system for our High Content phenotypic analysis. For the analysis of the results we use a dedicated script (the Harmony software) that calculates, in each GFP-TFEB-positive cell, the GFP intensity measured in the nuclear region, which is defined as the “DAPI”-positive area, and the GFP intensity measured in the cytosolic region, which is defined as a “ring” drawn by the software in the perinuclear region of the cytosol. This “ring” is drawn in such a way that its area and its distance from the nucleus is identical in each cell. Thus, the “cytosolic” TFEB intensity signal calculated by the OPERA system does not represent the exact amount of TFEB molecules in the whole cytosol, which could only be calculated by analysing the whole cytosolic volume and, therefore, by performing a 3D reconstruction of the cell (not doable with the OPERA system). Therefore, the nucleo-cytosolic ratio provided by the OPERA system is an indicative, rather than an exact, ratio between the nuclear and the cytosolic TFEB signal intensity. This ratio, however, is extremely reproducible in different experimental assays performed in several years (Settembre et al. EMBO J, 2012; Medina et al. Nat Cell Biol 2015; the present study) and reflects the nucleo-cytoplasmic distribution of TFEB observed by IF in different experimental conditions. The OPERA system, however, allows the quantification of thousands different cells simultaneously (about 2000 cells per condition were quantified in the experiment in Figure 1C), which makes our analysis highly unbiased and significant. We hope this clarifies the reviewer's point.

Is there any other potential mechanism that regulates the localization of TFEB?

Certainly “other potential mechanisms” involving either transcriptional or diverse post-translational TFEB modifications (e.g. acetylation, sumoylation, ubiquitination) may also be important for the modulation of TFEB subcellular localization. Such mechanisms, however, are out of the scope of our study, which describes S138 and S142 TFEB phosphorylation as a major mechanism modulating TFEB nuclear export.

Secondly, as the reviewer 3 mentioned that XPO1/CRM1-dependent nuclear export of TFEB has been reported in Cell Rep. 2018 May 15, therefore to know the TFEB-CRM1

direct interaction is indispensable for this manuscript. Authors replied in their rebuttal that they are unable to perform this biochemical experiment and that answer is unacceptable.

As explained earlier, proof of direct interaction between TFEB and CRM1 was already provided in a previous report (Kirli et al 2015). This is explained in the results and discussion sections.

Reviewer #3 (Remarks to the Author):

The authors' responses are acceptable and the revised paper has been improved. This reviewer appreciated the authors for sharing interesting unpublished observations.

We would like to thank the reviewer for positively evaluating our work.